# Modeling Biomass for Natural Subtropical Secondary Forest using Multi-Source Data and Different Regression Models in Huangfu Mountain, China

**Congfang Liu** [1,2], **Donghua Chen** [1,3,*], **Chen Zou** [3], **Saisai Liu** [1], **Hu Li** [3], **Zhihong Liu** [4], **Wutao Feng** [4], **Naiming Zhang** [1] and **Lizao Ye** [1]

[1] College of Computer and Information Engineering, Chuzhou University, Chuzhou 239000, China
[2] School of Civil Engineering, Nanning College of Technology, Nanning 530031, China
[3] School of Geography and Tourism, Anhui Normal University, Wuhu 241002, China
[4] College of Geographical Sciences and Tourism, Xinjiang Normal University, Utumqi 830054, China
[*] Correspondence: chendonghua@chzu.edu.cn; Tel.: +86-055-0351-0251

**Abstract:** Forest biomass estimation is an important parameter for calculating forest carbon storage, which is of great significance for formulating carbon-neutral strategies and forest resource management measures. We aimed at solving the problems of low estimation accuracy of forest biomass with complex canopy structure and high canopy density, and large differences in the estimation results of the same estimation model under complex forest conditions. The Huangfu Mountain Forest Farm in Chuzhou City was used as the research area. As predictors, we used Gaofen-1(GF-1) and Gaofen(GF-6) satellite high-resolution imaging satellite data, combined with digital elevation model (DEM) and forest resource data. Multiple stepwise regression, BP neural network and random forest estimation models were used to construct a natural subtropical secondary forest biomass estimation model with complex canopy structure and high canopy closure. We extracted image information as modeling factors, established multiple stepwise regression models of different tree types with a single data source and a comprehensive data source and determined the optimal modeling factors. On this basis, the BP neural network and random forest biomass estimation model were established for *Pinus massoniana*, *Pinus elliottii*, *Quercus acutissima* and mixed forests, with the coefficient of determination ($R^2$) and root mean square error (RMSE) as the judgment indices. The results show that the random forest model had the best biomass estimation effect among different forest types. The $R^2$ of *Quercus acutissima* was the highest, reaching 0.926, but the RMSE was 11.658 t/hm². The $R^2$ values of *Pinus massoniana* and mixed forest were 0.912 and 0.904, respectively. The RMSE reached 10.521 t/hm² and 6.765 t/hm², respectively; the worst result was the estimation result of *Pinus elliottii*, with an $R^2$ of 0.879 and an RMSE of 14.721 t/hm². The estimation result of the BP neural network was second only to that of the random forest model in the four forest types. From high precision to low precision, the order was *Quercus acutissima*, *Pinus massoniana*, mixed forest and *Pinus elliottii*, with $R^2$s of 0.897, 0.877, 0.825 and 0.753 and RMSEs of 17.899 t/hm², 10.168 t/hm², 18.641 t/hm² and 20.419 t/hm², respectively. In this experiment, the worst biomass estimation performance was seen for multiple stepwise regression, which ranked the species in the order of *Quercus acutissima*, *Pinus massoniana*, mixed forest and *Pinus elliottii*, with $R^2$s of 0.658, 0.622, 0.528 and 0.379 and RMSEs of 29.807 t/hm², 16.291 t/hm², 28.011 t/hm² and 23.101 t/hm², respectively. In conclusion, GF-1 and GF-6 combined with data and a random forest algorithm can obtain the most accurate results in estimating the forest biomass of complex tree species. The random forest estimation model had a good performance in biomass estimation of primary secondary forest. High-resolution satellite data have great application potential in the field of forest parameter inversion.

**Keywords:** biomass; GF satellites; multiple stepwise regression; BP neutral network; random forest

## 1. Introduction

As the biggest carbon sink system in the terrestrial system, the forest ecosystem performs an important role in balancing regional ecological environments and the global carbon cycle [1,2]. The biomass of forest vegetation accounts for 80% of total biomass of terrestrial vegetation [3], and the forest biomass, as an important indicator for the calculation of the carbon sink and carbon cycle in terrestrial ecosystems, is of great significance in analysis of spatial distribution patterns and dynamic changes in carbon storage by forest vegetation [4]. Therefore, accurate estimation of biomass is particularly important in the quantization of carbon sink and carbon flux in a region and the calculation of regional carbon inventory, accounting for the international carbon reserve and mitigation of constantly increasing $CO_2$ pollution [5,6]. The traditional biomass acquisition method is mainly based on field measurement, which has a heavy workload, takes a long time and inflicts great damage to the ecosystem. Remote sensing technology has become an important means of forest parameter inversion and forest resource investigation due to its advantages, such as fast data updating, large coverage area and comparable ground data [7,8].

So far, many scholars have carried out exhaustive research on biomass inversion, and the characteristics of the development in this field in the last ten years are concluded to be as follows: firstly, the existing optical remote sensing, microwave radar, laser radar and hyperspectral data have been widely used in forest biomass estimation. Among optical remote sensing data, the vegetation indices are mainly based on the spectral reflectance of forest canopy, and the construction of textural features is utilized to further establish the relationship with biomass [9]; synthetic aperture radar (SAR) and light detection and ranging (LiDAR) data have unique advantages in the estimation of forest height and construction of the three-dimensional structure of forests [10]; and with high spectral resolution, hyperspectral data can be used for inversion with various vegetation biochemical parameters [11–13]. Secondly, inversion with integrated multi-source data has become a trend. The most commonly used multi-source data integration is the integration of optical remote sensing and LIDAR data and hyperspectral and high-spatial-resolution data [14,15]. Thirdly, the linear regression model and machine learning model have been adopted as the main estimation models [14]. The most widely used regression models include multiple stepwise regression, principal component regression and ridge regression models, and the support vector machine and random forest algorithm are the machine learning models with the best performances [16]. However, the performance of one estimation method varies greatly among different research objects and sites [17–18].

There are the following problems in the study of forest biomass: Firstly, there are few studies on the biomass of primary secondary forests in the middle latitudes [19-20]. Second, the application of Gaofen satellite data to forest biomass research is rare; the application of GF-6 satellite data in forestry estimation is especially insufficient. At the same time, the applicability of biomass estimation methods is not stable. In view of the above, this article considers the Huangfu Mountain Provincial Natural Reserve of Chuzhou City, Anhui Province, China, as the research area. Using GF-1 panchromatic multispectral sensor (PMS) images (GF-1 data) and GF-6 wide-field-of-view (WFV) images (GF-6 data) as the main data sources, combined with field survey data from the same period, the forest biomass estimation model of Huangfu Mountain is established. The terrain in the Huangfu Mountain forest region is complex and mostly consists of low mounts and hills. The forest coverage rate in this region is high, and under the strong interference of human beings and nature, the fragmentation of the forest is growing [20]. The band information, principal component information and texture features influenced by GF-1 and GF-6 remote sensing were extracted as feature factors. We analyzed the correlation between forest survey data and characteristic factors. A multiple stepwise regression model was constructed to analyze the influence of data sources on the model fitting effect and the degree of contribution of characteristic factors. The characteristic factors with a high contribution to the mixed forest model were selected to construct a nonlinear regression model for

estimating the biomass of mixed forest to explore the performances of each image datum and algorithm in biomass estimation of natural subtropical secondary forest with multiple tree species. This research could provide a reference for the estimation of forestry resources, support the service function of information from the Chinese satellite Gaofen and provide a reference for the subsequent application of GF-1 and GF-6 networking satellite data, so as to promote the application of data from the Gaofen series satellites in forestry management and production.

## 2. Materials and Methods

### 2.1. Study Area

Huangfu Mountain is located in the Jianghuai Watershed in the central part of East China, with geographic coordinates of 117°58′–118°03′E, 32°17′–32°25′N and a total area of 3551.5 km². It is a subtropical deciduous broad-leaved forest. Located on the northern edge of the warm temperate semi-humid monsoon climate zone, the altitude is 45–385 m, the annual average temperature is 14.3 °C, and the annual precipitation is 1018.6 mm. The forest coverage rate is as high as 96.1%. According to the 2018 survey data of the second type of small class, it is composed of coniferous forests, broad-leaved deciduous forests, mixed broad-leaved forests and mixed coniferous and broad-leaved forests, of which 96.2% are pure-forest single-tree species. The main coniferous tree species are *Pinus massoniana* and *Pinus elliottii*, and the main hard-leaved broad-leaved tree species is *Quercus acutissima*, which accounts for more than 78.2% of the total forest area. This is the most intact and largest original secondary forest landscape belt in the area between the Yangtze River and the Huai River.

### 2.2. Data

#### 2.2.1. Ground Survey Data

The 2017 and 2012 subcompartment survey data of the Huangfu Mountain Forest was analyzed, and by taking the dominant tree species (*Pinus massoniana*, *Pinus elliottii*, *Quercus acutissima*) as the main species, the sample points were set according to the site conditions and characteristic appearance of the forest. The field survey of plot was carried out in Huangfu Mountain Forest from September to October 2018, during which, with the subcompartment as the unit of plot, the arboreal forest was measured using the angle gauge layout planning method, and after excluding the trees with a diameter at breast height (DBH) less than 5.0 cm, the mean basal area of each subcompartment was calculated using the arithmetic averaging method. A total of 315 subcompartment plots were investigated, and the distribution of plots is as shown in Figure 1. The survey data include species composition, dominant tree species, average diameter at breast height, average height of tree, crown density, stand age, growing stock volume per hectare, slope, aspect, elevation and other basic information of the plot, as shown in Table 1.

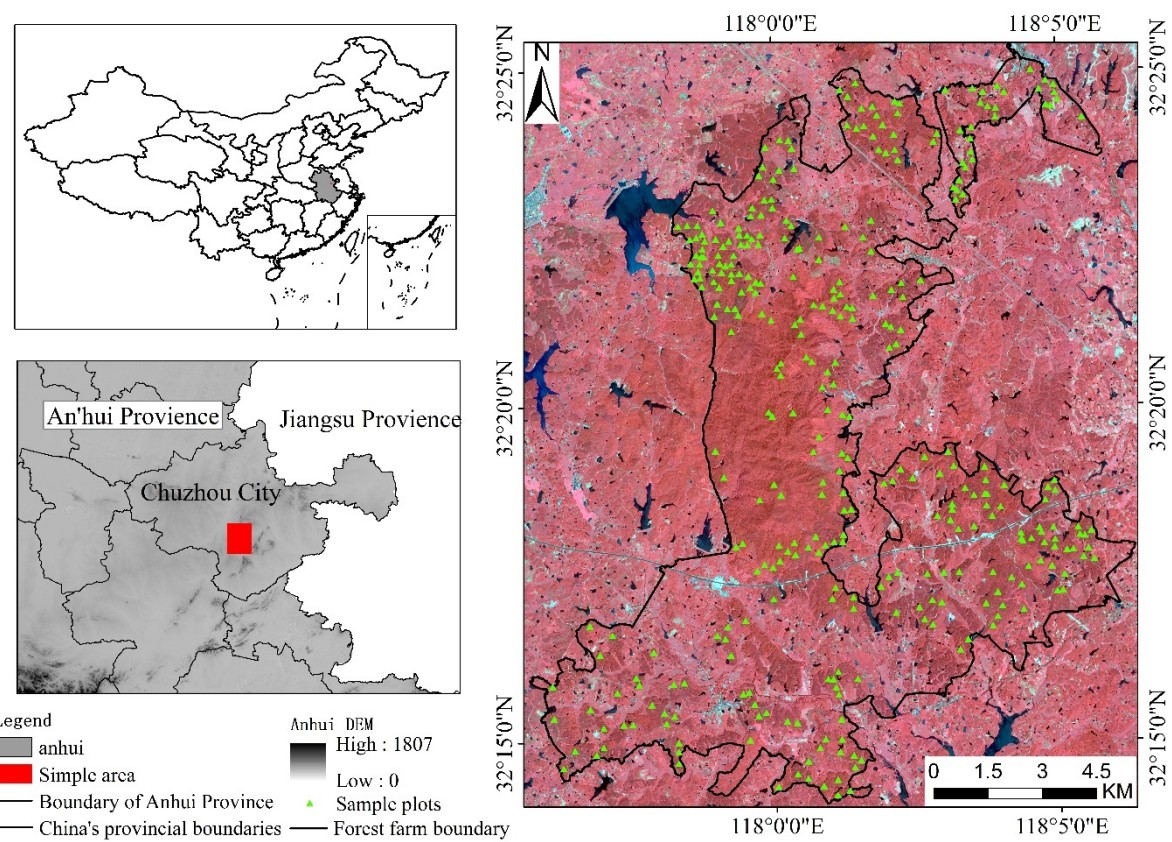

**Figure 1.** Standard false-color image of sample plot distribution.

**Table 1.** Basic Information on Forest Sample Plot Parameters.

| Types of Samples | Number of Samples | DBH/cm | | Height of Tree/m | | Trees/hm² | Crown Density | Forest Stock Volume /m³·hm⁻² | |
|---|---|---|---|---|---|---|---|---|---|
| | | Range | Average | Range | Average | Range | Average | Range | Range |
| Single species | 283 | 5.21–18.36 | 12.02 | 1.41–15.12 | 7.36 | 752–2951 | 1674 | 0.6–0.9 | 12.65–238.92 |
| *Pinus massoniana* | 123 | 6.21–18.28 | 13.28 | 5.86–13.12 | 7.42 | 752–2745 | 1821 | 0.8–0.9 | 30.64–234.15 |
| *Pinus elliottii* | 69 | 6.61–18.32 | 14.41 | 3.05–8.56 | 7.44 | 825–2350 | 1706 | 0.8–0.9 | 25.53–226.57 |
| *Quercus acutissima* | 52 | 6.12–15.22 | 9.01 | 2.62–9.56 | 5.52 | 725–2851 | 1224 | 0.9 | 21.86–234.94 |
| Broadleaf forest | 22 | 5.15–15.20 | 8.41 | 4.56–9.75 | 4.14 | 752–2125 | 1114 | 0.8–0.9 | 14.49–220.36 |
| Coniferous forest | 17 | 6.15–18.36 | 14.68 | 4.52–8.56 | 7.81 | 752–2359 | 1350 | 0.8–0.9 | 12.65–185.88 |
| Needle and broad-leaved mixed forest | 32 | 6.15–16.16 | 10.28 | 3.21–8.56 | 4.92 | 1440–2550 | 1241 | 0.7–0.9 | 27.23–170.71 |

To carry out the inversion of the forest biomass (Bio) of the sensing field, the biomass of the plots under monitoring were converted, and the vegetation information of the research area was analyzed. DBH and tree height were converted into forest growing stock volume according to the binary standing volume table. The biomass of each tree species was calculated using the parameters in the conversion equation of forest stock volume and forest biomass proposed by Fang et al.[21] (In this research, biomass refers to the biomass of arbor, excluding the biomass of shrub and herbs.) The regression equation is as follows:

$$B = aV + b \qquad (1)$$

In Equation (1), *B* refers to the biomass per unit area (t/hm²); *V* refers to the growing stock volume per unit area (m³/hm²); a and b are parameters. The estimation model was selected according to the stand type, and the corresponding values of *a*, *b* are as shown in Table 2.

**Table 2.** Model Parameters for AGB-Growing Stock Volume.

| Forest Types | a | b |
|---|---|---|
| *Pinus massoniana* | 0.51 | 1.0451 |
| *Pinus elliottii* | 0.5894 | 24.5151 |
| *Quercus acutissima* | 1.3288 | −3.8999 |
| Broadleaf forest | 0.8392 | 5.4157 |
| Coniferous forest | 0.5168 | 33.2378 |
| Needle and broad-leaved mixed forest | 0.7143 | 16.9654 |

2.2.2. Remote Sensing Image

In this article, 1 scene of GF-1 image data from 4 September 2018 and 1 scene of GF-6 image data from 28 October 2018 (with daily cloud cover of less than 5%) were selected for research. The GF-1 data, for which the width of a single scene is 60 km, include the image data of a panchromatic camera and multispectral camera, where the spatial resolution of the panchromatic band is 2 m, and the spatial resolution of the multispectral band is 8 m. The width of a single scene of GF-6 data is 80 km, and the spatial resolution is 16 m. In addition to the traditional multispectral band (B1–B4: 0.45–0.89), GF-6 data also carry red-edge I (B5: 0.69–0.73), red-edge II (B6: 0.73–0.77) and purple-edge bands (B7: 0.40–0.45) and yellow-edge bands (B8: 0.59–0.63). Both the images covered the research area completely, and the detailed parameters are shown in Table 3.

**Table 3.** Basic information of remote sensing images.

| Sensor type | GF-1 PMS | GF-6 WFV |
|---|---|---|
| Sensor Altitude/km | | 645 |
| Spectral range/μm | Multispectral | B1:0.45–0.52 |
| | | B2:0.52–0.59 |
| | | B3:0.63–0.69 |
| | | B4:0.77–0.89 |
| | | B5:0.69–0.73 |
| | | B6:0.73–0.77 |
| | | B7:0.40–0.45 |
| | | B8:0.59–0.63 |
| | Panchromatic 0.45–0.90 | |
| Pixel Size/m | 2 | 16 |
| Width/km | 60 | 864.2 |
| Revisit Cycle | 4 | 2 (Networking with GF-1) |

The images were preprocessed by using the ENVI software. The radiance calibration and atmospheric correction were carried out according to the absolute radiance calibration coefficient and spectral response function of the GF-1 PMS1 sensor and GF-6 WFV sensor released by the China Center for Resources Satellites Data and Application. In addition, the rational polynomial coefficient (RPC) was found using non-offset Google Maps and 30 m-resolution DEM data with the error being kept below one pixel. The resolution of corrected GF-6 data was 16 m. Then, the data of the multispectral band of 8 m and

panchromatic band of 2 m of the corrected GF-1 image were integrated by using the pan-sharpening method, and the spatial resolution of the integrated image was 2 m.

The auxiliary data include terrain data (DEM). The 1:30 m DEM data were adopted to extract the information on the elevation, slope and aspect of the research area, which were used as the terrain factors during model construction.

### 2.3. Feature Extraction

#### 2.3.1. Extraction of Spectral Information

The single-band information in the GF-1 image and GF-6 image was adopted as the spectral information. In addition to the 4 bands involved in the GF-1 image data, 8 bands of GF-6 image data were added (the information of 12 bands was adopted) to highlight the characteristics of each tree species on the basis of the original spectral information of surface features. Meanwhile, 7 common vegetation indices were added, including the ratio vegetation index (RVI), normalized difference vegetation index (NDVI), enhanced vegetation index (EVI), difference vegetation index (DVI), soil-adjusted vegetation index (SAVI), green normalized difference vegetation index (GNDVI) and infrared percentage vegetation index (IPVI). In addition, the MERIS terrestrial chlorophyll index (MTCI) and normalized difference red-edge version 1 (NDRE1) for GF-6 new bands were introduced for biomass estimation. The calculation equations are as follows:

$$MTCI = (B6 - B5) / (B5 - B3) \tag{2}$$

$$NDRE1 = (B6 - B5) / (B6 + B5) \tag{3}$$

where: $B3$, $B5$ and $B6$ refers to the spectral eigenvalue of the red band, red-edge band 1 and red-edge band 2 of the GF6 WFV image. *MTCI* is sensitive to the chlorophyll in the leaves of plants, and its ratio is proportional to the content of chlorophyll. *NDRE1*, with the wave crest and trough of red edge replacing the red band and near-infrared band used in NDVI, is usually used for vegetation classification and estimation of leaf area index [22].

#### 2.3.2. Extraction of Principal Component Characteristics

In principal component analysis, a set of variables that may be correlated were converted to a set of linearly independent variables through orthogonal transformation to achieve the purpose of dimensionality reduction. Principal component analysis was carried out for each band of GF-1 and GF-6 data, respectively, to achieve the purpose of isolating noise between bands and reducing the dimension of the data set. The results show that the accumulative contribution rate of the first principal component and the second principal component of the GF-1 image can reach 99.06%, and the contribution rate of the first principal component of the GF-6 image can reach 86.51%. Therefore, the first principal component (PCA1GF-1) and the second principal component (PCA2 GF-1) of the GF-1 image and the first principal component of the GF-6 image (PCA1 GF-6) were selected as the modeling factors.

#### 2.3.3. Extraction of Textural Features

Many scholars have proved that the addition of a textural feature could effectively improve the estimation accuracy of a model. A gray-level co-occurrence matrix based on a second-order statistical filter was adopted to extract textural features in 5 windows for data of GF-6 new bands and GF-1 bands. The commonly used textural features were extracted, including mean, variance, homogeneity, contrast, dissimilarity, entropy, second moment and correlation. The extraction windows were 3 × 3, 5 × 5, 7 × 7, 9 × 9, 11 × 11 and 13 × 13, with the direction of 0°, 90°, 45° 135°. According to the analysis, we finally decided to set the direction as 135°, and the step size as 1.

### 2.4. Multiple Stepwise Regression Model

Stepwise regression is a linear regression model method for selecting independent variables. The basic idea is to introduce the variables one by one, and the introduction condition is to have significant experience of partial regression sum of squares. Each time a new variable is introduced during model construction, the variables that have been selected by the regression model are tested one by one, and the variables deemed insignificant by the test are deleted to ensure that each variable in the set of independent variables is significant. Repeat this process until no more new variables can be introduced [22].

### 2.5. Machine Learning Algorithm

Two machine learning algorithms, BP neural network and random forest, were used to construct the forest biomass model of the mixed forest. The BP neural network algorithm is a multi-layer pre-feedback neural network algorithm for error back propagation, and it is currently the most widely used neural network algorithm. The BP neural network algorithm is divided into two parts: one is the signal forward propagation, that is, the calculation is carried out from the input to the output direction, and the other is the error feedback, that is, the correction of the weight and the threshold is carried out from the output to the input direction. The structure is divided into three parts: input layer, hidden layer and output layer[23-24].

The random forest algorithm is a type of machine learning algorithm. The algorithm is based on a decision tree and uses multiple classifiers to vote to determine the final result. It has the characteristics of high accuracy and strong resistance to overfitting. Random forest regression involves the establishment of multiple independent decision trees, random sampling of samples with replacement, model training according to the samples and obtainment of the integrated results of average or majority voting to avoid possible deviations due to the results of a single decision tree. Then, better estimation results can be obtained through characteristic factors.

The accuracy of each estimation equation was evaluated by taking the determination coefficient $R^2$ and root mean square error (RMSE) as the evaluation indices. The equations are as shown below:

$$R^2 = 1 - \frac{\sum_{i}^{n} (y_i - \hat{y}_i)^2}{\sum_{i}^{n} (y_i - \overline{y}_i)^2} \tag{4}$$

$$RMSE = \sqrt{\frac{\sum_{i}^{n} (\hat{y}_i - y_i)^2}{n}} \tag{5}$$

In Equations (3) and (4), $y_i$ refers to the measured value of biomass in the sample plot, $\hat{y}_i$ refers to the predicted value of biomass in the sample plot, $\overline{y}_i$ refers to the mean value of biomass in the sample plot and $n$ refers to the number of sample plots.

## 3. Results

### 3.1. Data Statistics and Processing

The data of plots were processed to exclude the trees with DBHs less than 5 cm and heights less than 1.5 m. The biomass data were analyzed by tree species, and on the basis of classification by stand age, the outliers were excluded according to the principle of triple standard deviation. Finally, 295 copies of data were selected for modeling, including 126 samples of *Pinus massoniana* (dominant tree species), 69 samples of *Pinus elliottii*, 52 samples of *Quercus acutissima* and 48 samples of broad-leaved forest and mixed broadleaf–

conifer forest. It should be noted that single-species biomass modeling was carried out using single-species data, such as *Pinus massoniana*, *Pinus elliottii* or *Quercus acutissima*. The principle of three standard deviations was applied to the data processing, the highest values in the measured data were removed and the biomass model of mixed forest species was established with the remaining survey data of all forest types. A total of 70% (204 samples) of the sample data were randomly selected as the training set for model construction, and the remaining 30% (85 samples) were used as the training set for model accuracy testing.

### 3.2. Determination of Feature Factors

The selection of feature factors is decisive for the accuracy of model. It will not only reduce the computation efficiency of the model, but also cause serious deviation in predicting outcomes if too many feature factors are introduced, or if the correlation between the feature factor and biomass is poor. A total of 412 feature factors were selected, including remote sensing band data (GF1: B1–B4; GF6: B1–B8), vegetation index data (16 copies in total), principal component analysis data and texture characteristic data, The feature factors are outnumbering the modeling samples (206). However, introducing all factors for model construction would raise the risk of overfitting; that is, the feature factors fit well, but the application effect of verification samples is poor, and thus the feature selection of variables would be conducted before modeling. The Pearson coefficient was adopted to analyze the correlation between feature factors and biomass to reduce the dimension of modeling factors. The optimal variables and correlation coefficients of biomass inversion models for four types of forests are shown in Table 4. As shown in the table, the mixed forest biomass correlated most to band 2 of the GF-1 image, with a correlation coefficient of −0.415. The next one was pca1, with a correlation coefficient of 0.236, and it was followed by three vegetation indices, i.e., ipvi, evi and dvi, for which the correlation coefficients were 0.293, 0.251 and 0.222, respectively. The correlation coefficient between the biomass of the mixed forest and the red-edge vegetation index of the GF-6 image reached 0.548, the correlation coefficient between it and B5_mean_13 × 13 was −0.262 and the correlation coefficient between it and new red-edge band vegetation index NDRE1 was −0.207. The correlation coefficients between the biomass of *Pinus massoniana* and PCA1$_{GF-1}$, band2 and b1_mean_3 × 3 of the GF-1 image were 0.318, 0.314 and 0.310, respectively, and the correlation coefficients between it and the vegetation index rvi and evi were 0.280 and 0.277, respectively; the correlation coefficient between the biomass of *Pinus massoniana* and the first principal component of the GF-6 image was 0.385, and the correlation coefficient between it and BAND6 was 0.366. The biomass of *Pinus elliottii* had extremely significant correlation with the normalized differential vegetation index (NDVI) and band 4, where the correlation coefficients between it and the vegetation index of GF-1 and GF-6 were 0.503 and 0.307, respectively, and the correlation coefficients between it and band 4 of GF-1 and GF-6 were 0.522 and 0.380, respectively. In addition, the biomass of *Pinus elliottii* was also significantly correlated to PCA1 GF-6, with a correlation coefficient of 0.360. The biomass of *Quercus acutissima* had a significant negative correlation with GF-1 band2 with a correlation coefficient of 0.675, as well as the b3_mean_3 × 3, with a correlation coefficient of 0.525. The biomass of *Quercus acutissima* had an extremely significant correlation with the GF-6 new red-edge index MTCI and green normalized differential vegetation index GNDVI of GF-6, with correlation coefficients of 0.509 and 0.396; it had a significant negative correlation with BAND6, with a correlation coefficient of 0.446. The biomass of mixed forest, *Pinus massoniana* and *Quercus acutissima* was strongly correlated to the terrain factor, and the elevation (45–385 m) and slope of the research area had a significant influence on biomass. The correlation coefficients between the GF-6 new band, new red-edge vegetation index and biomass of mixed forest, *Pinus massoniana* and *Quercus acutissima*, were greater than 0.36.

**Table 4.** Correlation between each factor and biomass [1].

|  | Mixed Forest | Correlation | *Pinus massoniana* | Correlation | *Pinus elliottii* | Correlation | *Quercus acutissima* | Correlation |
|---|---|---|---|---|---|---|---|---|
| GF-1 | PCA1$_{GF-1}$ | 0.236 ** | PCA1$_{GF-1}$ | 0.318 ** | ndvi | 0.503 ** | band2 | −0.675 ** |
|  | band2 | −0.415 ** | band2 | −0.314 ** | band 4 | 0.522 ** | b3_mean_3 | −0.525 ** |
|  | ipvi | 0.293 ** | rvi | 0.280 ** |  |  |  |  |
|  | evi | −0.251 ** | evi | −0.277 ** |  |  |  |  |
|  | dvi | 0.222 ** | b1_mean_3 | −0.310 ** |  |  |  |  |
| Land type | SLOPE | 0.468 ** | SLOPE | 0.500 ** |  |  | SLOPE | 0.476 ** |
|  | DEM | 0.519 ** |  |  |  |  |  |  |
| GF-6 | MTCI | 0.548 ** | PCA1$_{GF-6}$ | −0.385 ** | NDVI | 0.372 ** | MTCI | 0.549 ** |
|  | B5_mean_13 | −0.262 ** | BAND6 | −0.366 ** | PCA1 $_{GF-6}$ | 0.360 ** | BAND6 | −0.446 ** |
|  | NDRE1 | −0.207 ** |  |  | BAND4 | 0.380 ** | GNDVI | 0.396 ** |
|  |  |  |  |  |  |  | PCA1$_{GF-6}$ | −0.513 ** |

[1] Lowercase letters represent GF-1 modeling factors, and uppercase letters represent GF-6 modeling factors. ** Significant correlation at 0.01 level.

### 3.3. Analysis of Modeling Results of Different Images

Multiple stepwise regression is one of the classical multiple regression analysis methods. The procedure of this method begins with introducing the independent variables verified through F statistics one by one; and in each process, after introducing the third variable, the variables introduced previously are verified using the backward method to exclude the insignificant variable and the variable without any influence on the equation; then, the cycle is repeated until no variable can be excluded and no new variable can be introduced. In this article, the multiple stepwise regression model was adopted to build the models of three main tree species and mixed forest with the GF-1 data, GF-6 data and integrated data. The SPSS was applied to build a multiple stepwise linear regression model to conduct further screening on the factors being introduced into the models in combination with collinearity diagnostics, so as to ensure the accuracy of the regression model.

The $R^2$, RMSE (t/hm²) and F values of fitted regression equations of biomass models for four types of forests (mixed forest, *Pinus massoniana*, *Pinus elliottii* and *Quercus acutissima*) are as shown in Table 5. The biomass of each type of forest was calculated by utilizing the biomass regression model. As shown in Table 5, the biomass was extremely significantly correlated to modeling factors in the regression analysis, and the significance test probability F was 0.000 (<0.01). The best goodness-of-fit $R^2$s of the four tree species were 0.529, 0.622, 0.379 and 0.658, respectively, all of which were the fitting result of GF-1 combined with the GF-6 data model; furthermore, the modeling results of mixed forest, *Pinus massoniana* and *Pinus elliottii* with GF-1 data were better than those with GF-6 data, where the goodness of fit of the modeling result with GF-1 data was 0.44, 0.55 and 0.365, respectively, and the goodness of fit of the modeling result with GF-6 data was 0.436, 0.388 and 0.3, respectively. The modeling result of *Quercus acutissima* with GF-6 data was better than that with GF-1 data, with $R^2$s of 0.613 and 0.540, respectively. According to the modeling results, the GF-6 red-edge index MTCI plays an important role in model construction for mixed forest and *Quercus acutissima*, but it has a low interpretability in model construction for *Pinus massoniana* and *Pinus elliottii*.

**Table 5.** Multiple stepwise regression modeling results for different tree species.

|  | Mixed forest | $R^2$ | RSME/(t·hm²) | F | Sig |
|---|---|---|---|---|---|
| GF-1 | Bio = 0.103 × DEM + 0.349 × pca1 − 1.118 × band2 − 2569.065 × ipvi + 3259.183 | 0.444 | 30.42 | 68.023 | 0.000 |
| GF-6 | Bio = 449.076 × MTCI + 0.238 × DEM − 12.195 × B5_mea_13 +2.139 × SLOPE + 428.004 | 0.436 | 29.50 | 58.091 | 0.000 |

| | | | | | |
|---|---|---|---|---|---|
| GF-1&GF-6 | Bio = 926.384 × MTCI + 0.331 × DEM + 450.665 × NDRE1 + 707.176 × evi + 2.431 × SLOPE + 0.23 × dvi + 573.945 | 0.529 | 28.01 | 53.871 | 0.000 |
| ***Pinus massoniana*** | | | | | |
| GF-1 | Bio = 3.014 × SLOPE + 0.303 × pca1 − 0.777 × band2 − 99.567 × rvi + 1248.711 | 0.550 | 15.24 | 36.985 | 0.000 |
| GF-6 | Bio = −5.256 × SLOPE − 0.045 × PCA1 − 0.043 × BAND6 + 31.638 | 0.388 | 27.74 | 25.787 | 0.000 |
| GF-1&GF-6 | Bio = 3.656 × SLOPE − 0.024 × BAND6 − 0.043 × PCA1$_{GF-6}$ + 0.313 × PCA1$_{GF-1}$ + 562.712 × evi − 18.887 × b1_mea_3 + 403.054 | 0.622 | 16.29 | 32.571 | 0.000 |
| ***Pinus elliottii*** | | | | | |
| GF-1 | Bio = 0.076 × band4 + 225.897 × ndvi − 173.935 | 0.365 | 12.53 | 18.973 | 0.000 |
| GF-6 | Bio = 138.501 × NDVI + 0.03 × BAND4 + 0.026 × PCA1 − 26.226 | 0.300 | 13.23 | 9.415 | 0.000 |
| GF-1&GF-6 | Bio = 0.067 × BAND4 + 143.447 × ndvi + 0.027 × band 4 − 149.516 | 0.379 | 13.10 | 13.428 | 0.000 |
| ***Quercus acutissima*** | | | | | |
| GF-1 | Bio = 6.48 × SLOPE − 1.494 × band2 + 89.935 × b3_mea_3 + 1282.171 | 0.540 | 42.17 | 18.813 | 0.000 |
| GF-6 | Bio = 921.562 × MTCI − 22.662 × PCA1 + 1183.028 × GNDVI + 229.813 | 0.613 | 33.34 | 26.326 | 0.000 |
| GF-1&GF-6 | Bio = 728.141 × MTCI − 0.4 × band2 − 0.125 × BAND6 + 879.741 × GNDVI + 649.046 | 0.658 | 29.81 | 21.052 | 0.000 |

### 3.4. Modeling Analysis on Different Models for Mixed Forest

According to the multiple stepwise regression model, the accuracy of forest biomass estimation was low. Therefore, the BP neural network model and random forest algorithm model were introduced to explore the estimation accuracy of machine learning for different forest models and to find the most suitable biomass estimation model. Based on the correlation between modeling factors and biomass, the first six variables with significant correlation (DEM, SLOPE, MTCI, NDRE1, evi, dvi) were selected as the modeling factors for whole-forest biomass after removing the factors with strong collinearity. SLOPE, BAND6, PCA1$_{GF-6}$, PCA1$_{GF-1}$, evi and b1_mean_3 × 3 were selected as modeling factors for Pinus massoniana biomass. BAND4, ndvi and band 4 were the characteristic factors of the Pinus elliottii biomass model. MTCI, BAND2, BAND6 and GNDVI were the characteristic factors of Quercus acutissima biomass.

### 3.4.1. BP Neural Network

It is necessary to specify the quantity and structure of input and output sample sets prior to determining the structure parameters of a network, for it will make the training process complicated and cause the introduction of noise if there are too many input nodes, and sufficient information for modeling cannot be provided if the input nodes are insufficient. A BP neural network based on the Levenberg–Marquardt algorithm was constructed by using the neural network toolbox, with the number of nodes in the input layer being set as 6, the number of hidden neurons being set as 10 and the number of nodes in output layer being set as 1. Altogether, 70% of the data in the data sample set were used as training data, 15% of the data were used as verification data and 15% of the data were used as test data [25].

### 3.4.2. Random Forest Algorithm

The random forest algorithm has great robustness and higher accuracy, and it is suitable for the high-dimensional data sample set with significantly correlated features. The algorithm involves taking data at random with a replacement from the modeling data set using the bootstrap method repeatedly, and voting on the samples taken every time, and finally obtaining the prediction result through voting. The random forest model in the sklearn function of python was called, and 50 decision-making trees were built to finish the construction of the model through repeated training.

The modeling results show that better fitting results could be achieved when the BP neural network and random forest model were applied for biomass estimation. The accuracy of the verification was determined on reserved samples, along with the determination of the coefficient $R^2$ and RSME of the fitting effect, to verify whether the predicted

value of the model and actual observed value fit well. Figure 2 shows the fitting effect of the multiple stepwise regression equation, BP neural network and random forest model in terms of modeling samples and verification samples. As shown in the figure, the random forest model was used to estimate the forest biomass of the four forest types with the highest accuracy, and the biomass fitting degree of *Quercus acutissima* was the most accurate with an of $R^2$ of 0.926 and an RMSE of 11.658 t/hm². The fitting accuracy of *Pinus massoniana*, mixed forest and *Pinus elliottii* was 0.912, 0.904 and 0.879, and the RMSE was 6.765 t/hm², 10.521 t/hm² and 14.721 t/hm², respectively. This conclusion is consistent with the results of multiple stepwise regression, and the BP neural network also showed such results in the performance of these four forest types. In regard to the estimation effect of the three models, the random forest model was better than the BP neural network, and the multiple stepwise regression had the worst effect, which is consistent with the conclusion of Hu et al (2020) [23].

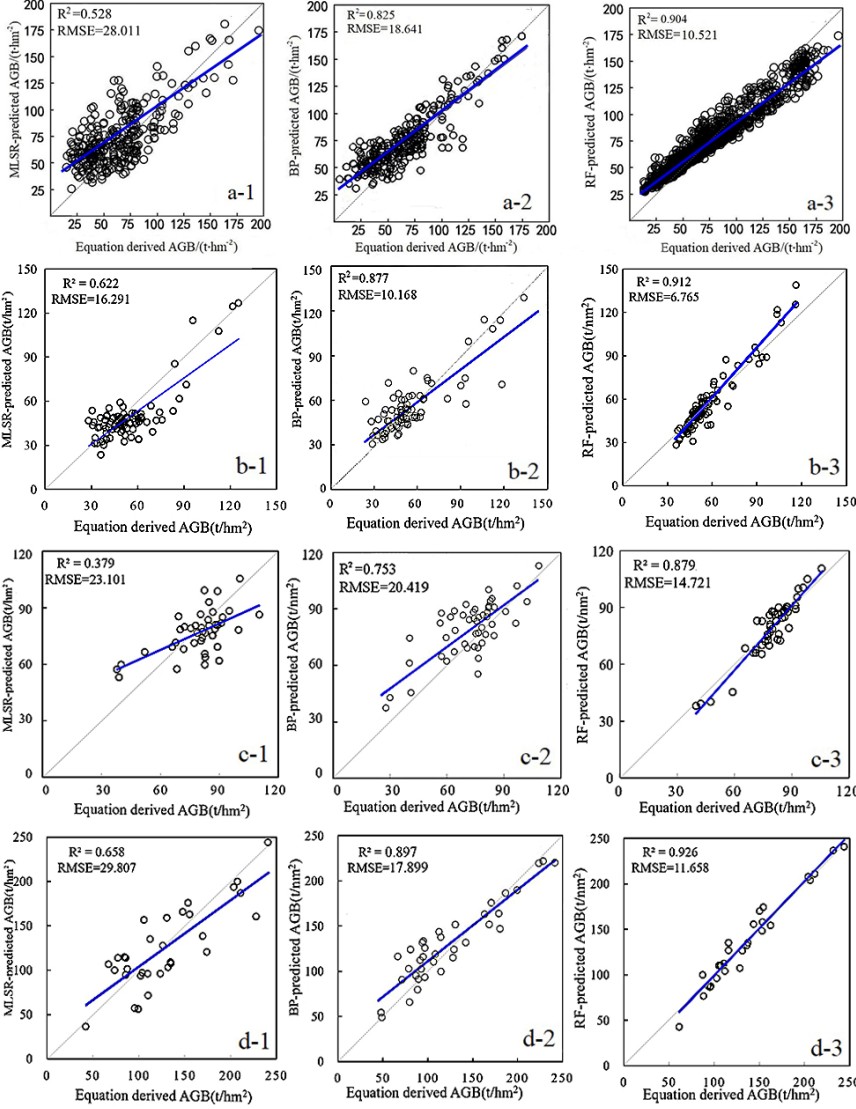

**Figure 2.** Comparison of measured and predicted biomass of four forest types using three estimation methods. (**a-1**–**a-3**) Biomass prediction results of mixed forest multiple stepwise regression, BP neural network and random forest models; (**b-1**–**b-3**) biomass estimation results of multiple stepwise regression, BP neural network and random forest models for *Pinus massoniana*; (**c-1**–**c-3**) biomass estimation results of multiple stepwise regression, BP neural network and random forest models for

*Pinus elliottii*; (**d-1**–**d-3**) biomass estimation results of multiple stepwise regression, BP neural network and random forest models for *Quercus acutissima.*

After three standard deviation screening sessions, the highest measured value of the mixed forest in this experiment was 196.575 t/hm², while the highest estimated value was 184.885 t/hm², which showed a certain phenomenon of underestimation of the high value. However, in the biomass estimation of *Pinus massoniana*, *Pinus elliottii* and *Quercus acutissima* single-tree species, the high estimation values of the machine learning model were more accurate.

The accuracy of the reserved samples was verified. The coefficient of determination ($R^2$) and root mean square error (RMSE) after fitting were used to verify whether the model was well-fitted and whether there was overfitting. The validation sample set was used for verification, and the results are shown in Figure 3. The model validation results of the four forest types are good. In the verification results, random forests and the performance of the multiple stepwise regression were more stable, and the $R^2$ and the RMSE were higher than forecasted for the model parameter; for example, the mixed multiple stepwise regression prediction of $R^2$ was 0.528, while the $R^2$ was verified as 0.562, and the RMSE values were higher, but the results of the validation may be due to the difference between the sample size, and thus it needs more tree sample verification.

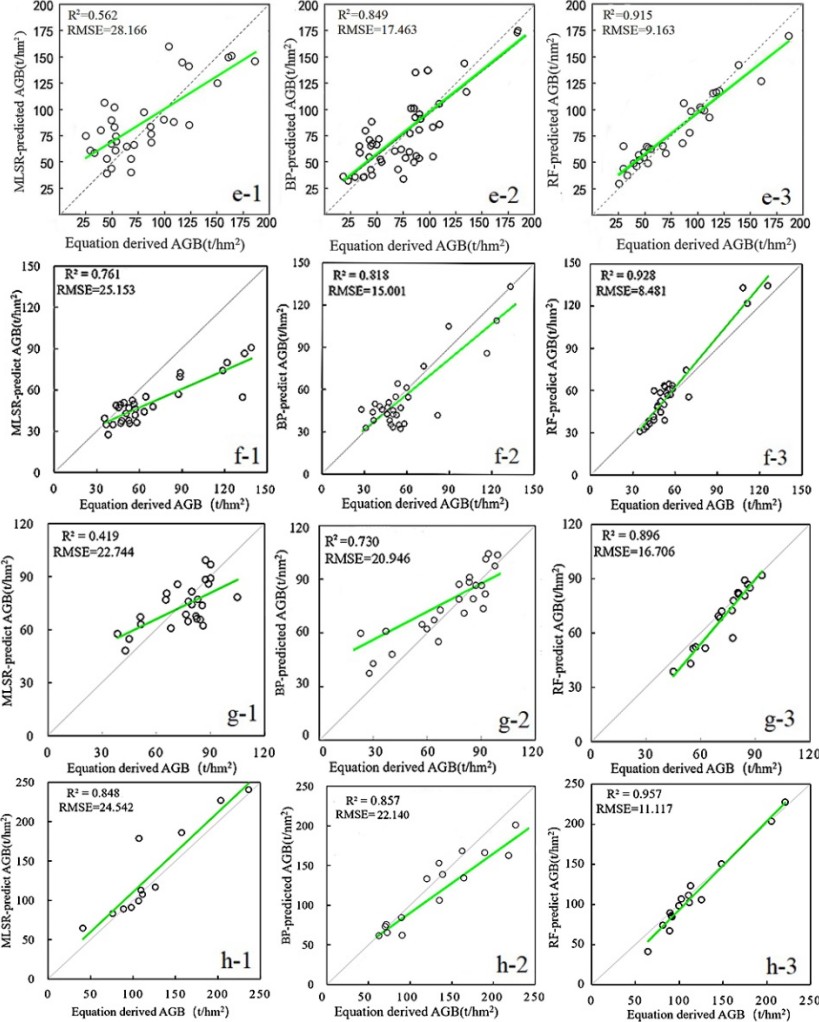

**Figure 3.** Comparison of biomass validation results of four forest types estimated using three estimation methods. (**e-1**–**e-3**) Biomass validation results of mixed forest multiple stepwise regression, BP neural network and random forest models; (**f-1**–**f-3**) biomass validation results of multiple stepwise regression, BP neural network and random forest models for *Pinus massoniana*; (**g-1**–**g-3**)

biomass validation results of multiple stepwise regression, BP neural network and random forest models for *Pinus elliottii*; (**h-1–h-3**) biomass validation results of multiple stepwise regression, BP neural network and random forest models for *Quercus acutissima.*

*3.5. Distribution Map of Biomass Estimation Results*

After estimating the four forest types, random forest was applied to estimate the biomass of different forest types. Taking mixed forest as the benchmark, a higher fitting accuracy was retained than for mixed forest, taking Masson pine and Quercus acutissima as examples. If the fitting accuracy was lower than that obtained for the mixed forest, the results of the mixed forest were retained.

Figure 4 shows the spatial distribution of forest biomass in Huangfushan, with biomass values ranging from 5.238 t/hm² to 344.039 t/hm². The difference between the three estimation models is obvious, and the difference was significant in the central forest area. The multiple stepwise regression model underestimated the high value in the central forest area. The random forest model had the best effect on the estimation of a high biomass value, with the highest biomass value reaching 344.039 t/hm².

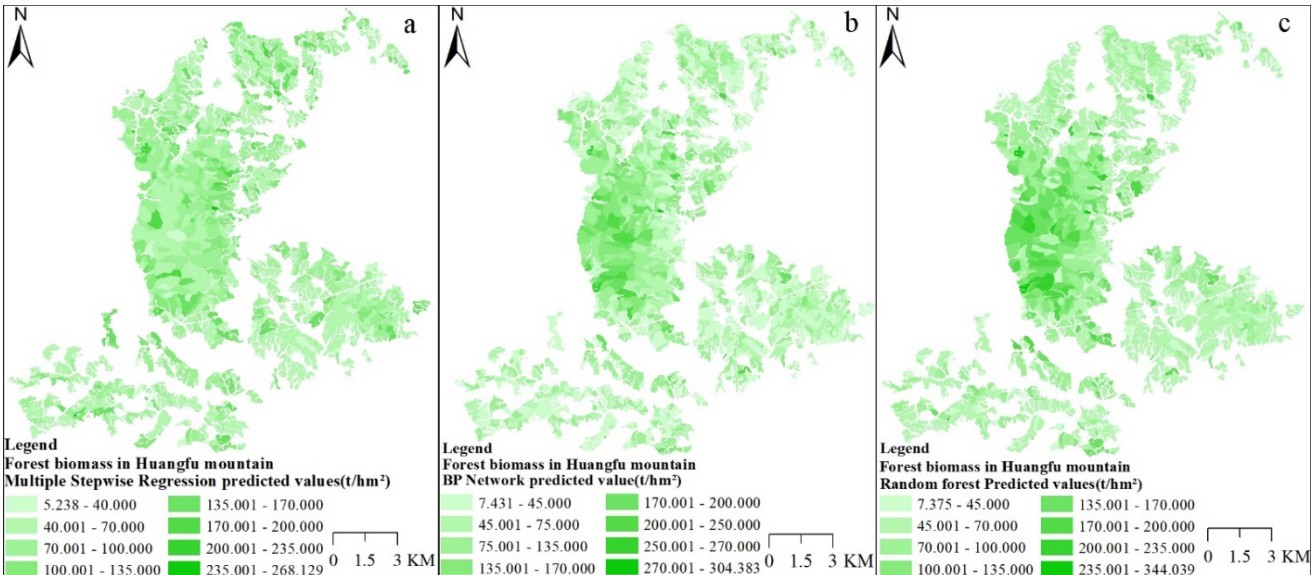

**Figure 4.** Distribution of biomass estimation results of different models. (**a**) Biomass estimation by multiple stepwise regression; (**b**) biomass estimation results of BP neural network; (**c**) random forest biomass estimation results.

The estimation results of biomass are shown in Table 6, and all models had certain errors. The multiple stepwise regression could not reach a high value of biomass estimation (239.19 t/hm² < 313.48 t/hm²). The estimation results of the random forest and BP neural network models were closest to the biomass range of forest resource data, but the overall estimation of biomass was too high. The average biomass of the random forest was higher than that of the BP neural network, which was higher than that of multiple stepwise regression, and the specific value was 65.115 t/ hm² < 56.456 t/hm² < 49.375 t/hm² < 47.482 t/hm². The random forest model was sorted from small to large standard deviation, and it was 32.530 t/hm². The BP neural network was 34.964 t/hm², and the multiple stepwise regression was 36.924 t/hm². In contrast, the random forest model had a lower degree of dispersion.

**Table 6.** Biomass estimation results statistics (t/hm²).

|  | Mean | Standard Deviation | Min | Max |
|---|---|---|---|---|
| Field survey data | 42.573 | 39.546 | 12.653 | 238.925 |
| Forestry resource data | 49.375 | 42.513 | 0 | 313.482 |

| | | | | |
|---|---|---|---|---|
| Multiple stepwise regression | 49.375 | 36.924 | 5.238 | 268.129 |
| BP neural network | 56.456 | 34.946 | 7.431 | 304.384 |
| Random forest model | 66.115 | 32.530 | 7.375 | 344.039 |

## 4. Discussion

In this paper, image information of GF-1 and GF-6 data, texture features and terrain data were extracted as feature factors, and the correlation between measured biomass and feature factors was analyzed. For mixed forest and broad-leaved forest, the MTCI constructed by the GF-6 red-edge index had the highest correlation with measured biomass, reaching 0.548 and 0.549, which is consistent with Jiang et al. (2021) [26].

In the multiple stepwise regression model, the results of different estimation data sources are very different. The joint data of GF-1 and GF-6 were the best, which is consistent with the results of Jiang et al. (2021). However, the coefficient of determination of coniferous forest fitting in this study was slightly lower than that found by Jiang et al. (2021), which is speculated to be due to the fact that this study area was a primary secondary forest, while Jiang et al.'s (2021) study area was a plantation [27].

Multiple stepwise regression, BP neural network and random forest models were used to estimate the biomass of mixed forest. The best estimation result was achieved with random forest, with an $R^2$ of 0.904. This conclusion is consistent with Ding et al.'s (2021) result, but the fitting accuracy is lower than Ding et al.'s (2021) fitting accuracy, which may be caused by the difference in data sources [28].

After constructing BP neural network and random forest models for three single-tree species, Masson pine, Prunus pine and Quercus acutissima, it was found that the random forest model still maintained a good estimation performance. The optimal estimation accuracy was found for *Quercus acutissima*, with an $R^2$ that reached 0.926, followed by *Pinus massoniana*, with an $R^2$ that reached 0.912. The estimation accuracy was higher than that of mixed forest ($R^2$ = 0.904), and the worst estimation effect was that of s *Pinus elliottii* ($R^2$ = 0.879). All the estimated results are up to standard. The reason for this result may be that broad-leaved oak species' ability to reflect electromagnetic radiation is higher than that of coniferous forests (*Pinus massoniana* and *Pinus elliottii*). The lowest estimation result for the *Pinus elliottii* species may be due to the small number of samples. We recommend increasing the sample size of the *Pinus elliottii* species in this area in future studies.

With the development of Chinese satellites, the quality and data of satellite data sources have been improved continuously, and data acquisition is no longer a challenge. Due to the strong development of computer technology and updated mathematical models, the accuracy of biomass estimation inversion has been further improved, and a trend of carrying out estimation inversion by utilizing multi-source data and neural networks and machine learning algorithms in the field of biomass estimation is gaining traction. In this article, the integrated GF-1 and GF-6 data and three estimation methods were studied, but the correlation between each characteristic factor and the accuracy of biomass estimation was only described qualitatively without quantitative evaluation. In addition, only one artificial neural network and one machine learning algorithm were discussed in this article. We shall carry out estimation based on multiple algorithms and functions to obtain better accuracy.

## 5. Conclusions

In this article, taking the Huangfu Mountain Forest as the research area, the image data of GF-1 and GF-6 satellites were processed, in combination with measuring biomass data of plots, extracting factors with extremely significant correlation using the method of Pearson correlation coefficient, based on which the multiple stepwise regression was constructed, to analyze the applicability and accuracy of mixed forest and single-tree species modeling with the integrated GF-1 and GF-6 image data. Then the BP neural network and

random forest model of mixed forest and single tree species were established, and the accuracy was verified. To analyze the most suitable biomass estimation model for different forest species.The results show that, with the information of three image bands (GF-1: band2; GF-6: BAND4; BAND6), two principal component factors (PCA1 GF-1; PCA1 GF-6), seven vegetation indices (GF-1: ipvi, evi, dvi, rvi, ndvi; GF-6: MTCI, NDRE1, NDVI, GNDVI), two terrain factors (DEM, SLOPE) and two textural feature factors (GF-1: b1_mea_3; GF-6: B5_mea_13) being introduced into model construction, a better fitting effect is achieved. In addition, the results of constructing biomass estimation models with different data sources by utilizing multiple stepwise regression show that the estimation accuracy of integrated data modeling is higher than that of single-source modeling; the estimation result of biomass modeling for *Pinus Massoniana* and *Pinus Elliottii* with GF-1 image data was better than that for biomass modeling with GF-6 image data, while the biomass estimation result of *Quercus Acutissima* achieved by using GF-6 image data was better than that achieved by using GF-1 image data. Based on the combined data of GF-1 and GF-6, the BP neural network and random forest biomass estimation model of different forest types were established, and the accuracy of the model was verified. The final results show that the random forest model had the highest biomass estimation accuracy in both single-tree species and mixed forest.

**Author Contributions:** Formal analysis, data processing, investigation, formal analysis, composition—review and editing, C.L.; guidance on methods, supervision, project management, D.C.; guidance on methods, H.L.; software assistance, N.Z., C.Z. and S.L.; acquisition of plot sources, first draft, W.F. and Z.L.; modifying the evaluation, L.Y. All authors have read and approved the version of the manuscript for publication. All authors have read and agreed to the published version of the manuscript.

**Funding:** This study was supported by the following projects: major project of high-resolution Earth Observation System (No.76-Y50G14-0038-22/23); Anhui Province key research and development plan (No.2021003, No.2022107020028, No.202104A07020002); Anhui Science and Technology Major Program (No. 202003A06020002); Anhui Province Special Support Program for Innovation and Entrepreneurship Leading Talents (2019); Anhui University Collaborative Innovation Project (No. Gxxt-2021-048); Chuzhou Science and Technology Project (No.2021ZD015); and Anhui Universities Outstanding Youth Research Projects.

**Institutional Review Board Statement:** Not applicable.

**Informed Consent Statement:** Not applicable.

**Data Availability Statement:** The processed data required to reproduce these findings cannot be shared at this time as the data also form part of an ongoing study.

**Acknowledgments:** We would like to express our sincere thanks to Chuzhou Forestry Bureau for providing forest resource data, and to Zhao Bo et al. from the Huangfu Mountain Provincial Natural Reserve for assisting in a large number of surface observation experiments.

**Conflicts of Interest:** The authors declare no conflicts of interest.

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
