# Peer review of "Modeling Biomass for Natural Subtropical Secondary Forest Using Multi-Source Data and Different Regression Models in Huangfu Mountain, China"

_sustainability, doi:10.3390/su142013006_

Round 1

Reviewer 1 Report

The authors constructed an aboveground biomass model by using GF satellites dataset and traditional/modern statistical methods (linear regression, BP (back propagation) and RF (random forest)) in this manuscript for Huangfushan forest. However, such work is not new but new for this study area. The dataset was a little small but suitable to the current analysis. More general comment is that the present of results.

On one hand, the authors found that the multiple stepwise regression model has the worst performance, but I didn’t think it is a conclusion because the comparison between linear regression models and machine learning methods is inappropriate. In linear regression, the linear relationship between dependent and independent variables were described and statistical features (significance, collinearity, etc…) must be strictly adhered to, whereas the BP and RF methods were based on the principles of classification regardless of linear/nonlinear relationship. In this sense, I suggested that removing multiple linear regression. On the other side, I noticed that there are four types of forests (multi-tree species (32 plots), Pinus massoniana (123 plots), Pinus elliottii (69 plots) and Quercus acutissima (52 plots)) in the used dataset and the authors only presented results for the first forest type. At present, did the aboveground biomass models for multi-tree species were constructed by 32 plots or all plot (315 plots)? Therefore, I am worrying about the accuracy of results based on such a sample size of the modeling data before a clear description for modeling data were added.

Below, I list my minor comments on the manuscript. The number indicates the rows of the manuscript.

Tittle: the title as it is is very confusing, which such name makes the title to give the impression that the tree-level biomass is estimated for some species.

Line 51: give the full name of SAR and LIDAR when the abbreviations appear at first time. The same to the following abbreviations (DBH, etc…)

Line 87: two expressions “Mount Huangfu”, “Huangfu Mountain” and “Huangfushan” should be unified

Lines 112-115: only the size, DBH, Height, stand density and crown density can be found on Table 1, such information of stand age, growing stock volume per hectare, slope, aspect, etc…were not shown in Table 1.

Table 1: “Hight” was incorrect, I guess it was “Height”

Table 1: how did you deal with the plots of broadleaf and coniferous forest? I am confused here is that the classification of forest types, which was closely related to the following modeling work.

Lines 196-202: Please add references to support your argument

Line 342: the determination coefficient and root-mean-square error were same across the sub-figures.

Line 366: “…above 200 t/ha2 and below 0 t/ha2 considered as overestimation or underestimation of the actual biomass…”, how did the authors conclude on this point of the number 200 or 0? Please give some supports.

Author Response

Dear Review Expert,

Hello, thank you for your valuable comments on this manuscript. No matter in the theoretical part, the structure of the paper and the small details, all the experts have given me very detailed opinions and guidance. Here, thank you very sincerely.

According to your Comments on the article, I have revised one by one. Please check Response to Reviewer Comments for details. I also marked the corresponding position in the paper in red/blue, please review.

Thank you again for your valuable comments. If there is still any deficiency in the review process, please inform us in time. I look forward to having more discussion with you. And hope the article can pass the review and publish as soon as possible. I wish you all the best.

Kind regards,

Ms. Liu Congfang

Reviewer 2 Report

In abstract: Change GF-1 and GF-2 by Gaofen-1 (GF-1) and Gaofen-2 (GF-2) High-resolution Imaging Satellite data.

In introduction: Change PMS by: Panchromatic and Multispectral Sensor (PMS) ; Change WFV by: wide field of view (WFV) sensor

In Materials and Methods:

I believe you mean: The main coniferous tree species are Pinus massoniana and Pinus elliottii  (line 95 and line 96)

All scientific names must be written in italics. For example: Quercus acutissima, Pinus massoniana and Pinus elliottii.

Change in line 165 NDRE1 by: Normalized Difference Red-edge Version 1 (NDRE1)

In line 224, correct (3) by (4); in line 225, correct (4) by (5); in line 226, correct: In Equation (3), (4) by In Equation (4), (5)

In Results:

Identify in figures 2 and 3 the models multiple stepwise regression, BP neural network and random forest

The values shown in lines 350-353 are different from figure 2. It is necessary to review the text and figure 2.

correct t/h2 by t/ha (line 366)

Author Response

Dear Review Expert,

Hello, first of all, thank you for your affirmation of this manuscript. It is you who have given me the confidence and courage to publish my first SCI paper. I would like to express my heartfelt gratitude to you. At the same time, thank you for your valuable advice, which makes me more careful about the details of the manuscript. Thank you very much again.

According to your Comments, I have made modifications one by one. Please check Response to Reviewer Comments for details. I also marked the corresponding position in the paper in red/blue, please review.

Thank you again for your valuable comments. If there are still some inadequacies in the review process of this paper, please inform us in time. I look forward to having more discussion with you. And hope the article can pass the review and publish as soon as possible. I wish you all the best. Kind regards, Ms. Liu Congfang

Reviewer 3 Report

This paper touches upon an important issue within the debate on the methods on biomass estimation and the accuracy of data.  The paper show an innovative method

 GF-1 and GF-6 combined data and 30 random forest algorithm that allow to product accurate data. This paper presents a revolutionized and innovative analysis.  Its shows that the random forest model is the best for mixed forest biomass estimation.

 The analytical framework and the theoretical scheme are absent, this have to be done. Analysis in the text is not consistent, also the organization of the manuscript need to be ameliorated

Before providing detailed comments to the specific sections, I have some general suggestions to strengthen the analytical consistency.

Overall comment

The authors need to reframe, the discussion and conclusion. If the introduction is not clear, there is still lacking a clear context or not well structured. The manuscript need an English editing to review all the text.

Conclusion is well written, more development can be done. The discussion section is not done in the right way, no comparison is done with other studies. This discussion does not rise weak side of the study. 

Detailed comment

Line 19 to 22: This sentence is not clear.  rephrase it

Line 22-27. this sentence is too long. rephrase this

Line 38.  "the biggest"?? perhaps  "one of the biggest"

Line 63: mi-latitude?? specify the region , because in Europe and America many research have been  published

Line 66-72.  too long , rephrase it

Line 84.  this research need an analytical and conceptual framework to be consistent on result

text:   an english review have to done in all the manuscript

figure 1: review the legend of this map.  they are many mistakes. Sample area??

line 103: check this date to be consistent

provide the source of all the table and figure

figure 2: state as figure 2 a, b, c, and explain it in the sections

figure 3 : the same Remark as figure 2

figure 4: specify the model by each biomass map. what is the unit?

discussion section: the discussion section is not consistent, there is not reference to others studies

All remarks and comments are in the manuscript.

 Hope these comments are helpful to improve the manuscript for submission in Sustainability journal

Author Response

Dear Review Expert,

Hello, first of all, thank you for your affirmation of this manuscript, and also thank you for your valuable comments, which made me more careful about the logical structure, theoretical framework and details of the manuscript. Thank you very much again.

According to your Comments, I have made modifications one by one. Please check Response to Reviewer Comments for details. I also marked the corresponding position in the paper in red/blue, please review.

Thank you again for your valuable comments. If there are still some inadequacies in the review process of this paper, please inform us in time. I look forward to having more discussion with you. And hope the article can pass the review and publish as soon as possible. I wish you all the best.

Kind regards,

Ms. Liu Congfang

Round 2

Reviewer 1 Report

Thank-you for your consideration of the recommendations. The revised manuscript has taken into consideration the minor points and the authors made an effort to answer or discuss the major points in great detail. However, the major point of the data of the multiple stepwise regression models for me still is not solved. I note just two things that should be dealt with:

1. what’s the definition of “Mixed forest” in the manuscript? If the three types of forests (Pinus massoniana (123 plots), Pinus elliottii (69 plots) and Quercus acutissima) were included in the so-called “Mixed forest”, I agree to accept this study after minor revision, such as adding the definition of “Mixed forest”. Otherwise, I would strongly suggest that developing biomass models for the other three types of forests by using BP and RF approaches. Actually, I don’t agree with the inference of Line 329-333 in the newly manuscript, which stated that the fitting accuracy of biomass model for mixed forests is low and therefore developing this model by machine learning approaches. In my mind, the models for other three types of forests didn’t provide a high fitting accuracy. Hence, it is necessary to do the biomass model for the other three forests types using BP and RF.

2. the “…in Huangfu Mountain, Anhui Province, China,in Multi-Source Data,and linear and Machine Learning Regression Models” in the newly title seems to be rambling, it may be better to change it to something like “Modeling biomass for natural subtropical secondary forest by multi-source data and different regression models in Huangfu Mountain, China” or “Developing biomass models for natural subtropical secondary forest using multi-source data and different regression methods — a case in Huangfu Mountain, China.”.

Author Response

Dear reviewer

Hello, thank you for your review of this hand, and I am deeply grateful for your valuable opinions on the structure and content of this paper. Your opinions make this paper more perfect, and also let me benefit a lot as a learner in the field of remote sensing. I would like to express my heartfelt thanks to you.

In view of the questions you raised, I have revised the text and marked it in red for your reference. Thank you again for your review. I wish you all the best.

Kind regards,

Ms. Liu Congfang

Reviewer 3 Report

in the discussion section: Jiang's study? you have to give the year.

the discussion still to be improve

Author Response

(The authors gave the same response as above.)
